# Quantum Circuit Compilation:
# An Emerging Application for Automated Reasoning

**Davide Venturelli[‡,*], Minh Do[†,**], Bryan O'Gorman[‡,×], Jeremy Frank[†], Eleanor Rieffel[†],**
**Kyle E. C. Booth[+], Thanh Nguyen[*], Parvathi Narayan[±], Sasha Nanda[⊤]**

[†]Planning and Scheduling Group, NASA ARC; [‡]Quantum Artificial Intelligence Laboratory, NASA ARC;
[*]USRA Research Institute for Advanced Computer Science, USA; [**]Stinger Ghaffarian Technologies, Inc., USA;
[+]Dept. of Mechanical & Industrial Eng., University of Toronto, Canada; [*]Computer Science Dept., Dartmouth College, USA;
[±]Computer Science Dept., Washington University in St. Louis, USA;
[⊤]Physics Department, Caltech, USA [×]Depts. of Chemistry and EECS, University of California, Berkeley, USA

## Abstract

Quantum computing is an information processing paradigm with the potential to solve certain problems faster than any algorithm running on classical computer architectures. In the next few years, new processors will be developed that support quantum computations exceeding the simulation ability of even the largest classical computer systems. A number of academic and industrial groups are developing prototypes of such devices, also known as NISQ (Noisy Intermediate Scale Quantum) processors. Much as software must be compiled to run on classical computers, quantum algorithms must be compiled to take into account the constraints of particular NISQ devices. Especially in these early prototypes, algorithm performance degrades with runtime due to noise; for this reason, minimizing the runtime of the compiled algorithm (which is represented by a "quantum circuit") is critical. We describe a software framework to enable an automated reasoning approach to Quantum Circuit Compilation for NISQ architectures (QCC-NISQ), and our current implementation of it as part of software suite for automated, architecture-aware, compilation for emerging quantum computers. The key components of this suite are a circuit synthesizer, a QCC solver, and a visualizer. These tools provide critical support for the continued development of practical quantum computers and research into quantum algorithms.

## 1   Introduction

Quantum computing is an emerging computational paradigm with the potential to solve certain problems faster than any algorithm running on classical computer architectures. The breadth of quantum computing applications will become clearer in the next few years as new processors are developed that support quantum computations exceeding the simulation ability of even the largest classical supercomputers. The emerging gate-model *noisy, intermediate scale quantum* (NISQ) processor units, currently in the prototype phase, are *universal* in that, once scaled up, they can run any quantum algorithm.

Much as software must be compiled to run on classical computers, quantum algorithms, also referred to as *logical quantum circuits*, must be compiled to take into account the constraints of particular NISQ devices. Especially in these early prototypes, algorithm performance degrades with runtime due to noise; for this reason, minimizing the runtime of the compiled circuit is critical. Current NISQ ar-

chitectures have geometric limitations (e.g., connectivity), the specifics of which vary from processor to processor. In this paper, we concentrate on the problem of producing optimally-compiled circuits given the geometric limitations of the processor. Variants of this problem are known as the qubit mapping, qubit routing, qubit allocation, and qubit movement problem. We refer to this problem as Quantum Circuit Compilation for NISQ architectures (QCC-NISQ).

We focus on solid-state architectures based on superconducting quantum bits (qubits), which are among the most advanced NISQ processors. As one example of a geometric limitation, the planar architecture of these processors means that quantum operations can be carried out only between nearest-neighbor locations (qubits). A variety of small superconducting processors, with varying architectures, already exist. Such processors include the 20-qubit IBMQ20 by IBM made available through the Q-Network (IBM Corp. 2018); the 8-, 16-, and 19-qubit chips by Rigetti Computing (Rigetti Computing 2018), and the 72-qubit Bristlecone processor unveiled recently by Google (Google AI Blog 2018). The 50-qubit Intel Tangle Lake chip (Intel Corp. 2018) and a new IBM 50-qubit device are under test and evaluation. Other commercial players in the superconducting NISQ race include Alibaba (Alibaba Cloud 2018) and Quantum Circuits Inc. (Ofek et al. 2016).

In this paper, we detail our software suite based around applying AI Planning, aided by Constraint Programming (CP), to solve the QCC-NISQ problem. Our suite targets: (1) multiple gate-model quantum computing hardware platforms built by different companies (specifically, at the time of writing, Google, Rigetti, and IBM), and (2) different combinatorial optimization problems that can be solved with specific quantum algorithms such as the quantum alternating operator ansatz algorithm (QAOA) (Hadfield et al. 2019) (e.g., Max-Cut, graph coloring, and job shop scheduling). While we have previously published technical details on how we model and solve the QCC-NISQ problem, in this paper we report not only on the actual QCC-NISQ solver, but also on our software suite in its entirety: its components, engineering, and deployment in this promising application area for AI technologies.

## 2 QCC for NISQ Devices

In the circuit model of quantum computation, a quantum algorithm is expressed conceptually as a *logical quantum circuit*, consisting of a series of quantum operations called *quantum logic gates*. *Quantum processors* are physical devices that implement these quantum logic gates so that the desired quantum operations can be carried out on the quantum states stored in the qubits. In simple cases, the quantum logic gates directly correspond to *physical* quantum gates on the quantum processor, but more typically the processor has physical constraints that prevent a quantum logic circuit, describing the desired algorithm, from being directly implemented.

At a high level, these constraints can be classified into two types: (1) gate set constraints (i.e., those that specify the set of logic gates the processor is capable of applying), and (2) geometric constraints (i.e., those that specify upon which sets of qubits the available logic gates can be applied, limited by, for example, processor connectivity). Although these constraints differ among quantum processors, quantum algorithms can be re-expressed respecting the processor constraints with polynomial overhead in the number of gates (Brierley 2017). As such, for theoretical algorithmic work, the design of logical quantum circuits without concern for the implementation constraints of physical devices is sufficient. However, to implement a quantum algorithm on an actual device, these constraints must be efficiently addressed to take full advantage of NISQ processors.

In this work, we focus on a particular approach to addressing geometric constraints associated with processor connectivity. The approach maps logical qubits to physical qubits on the processor and iteratively updates the mapping through the insertion of additional gates in the course of the computation so as to enable the logical operations to be implemented respecting the physical contraints. This problem is often referred to as "quantum compilation," though quantum compilation usually involves addressing gate set constraints as well as geometric (e.g., connectivity) constraints. Another simple constraint is that gates involving the same qubit cannot be executed in parallel. A generalization of this constraint is a "cross-talk" constraint that may prevent gates in physical proximity from being executed at the same time (Booth et al. 2018).

QCC-NISQ frequently requires adding supplementary operations supported by the hardware to those specified in the idealized circuit. Current superconducting quantum processors have planar architectures with connections only between *nearest-neighbor* locations (qubits), resulting in restrictions as to where gates can be applied. Specifically, a gate can operate only on qubit states located on adjacent qubits on the chip. To compensate for the nearest-neighbor limitation, *swap* gates can move qubit states between connected qubits to reach a configuration where the desired gate, specified in the idealized circuit, can be applied. Current quantum computational hardware suffers greatly from *decoherence* (akin to noise), which degrades the *fidelity* of the computation (Bishop 2017). In NISQ processors, decoherence is intimately linked to the duration of the executed circuit that carries out the quantum computation, so it is

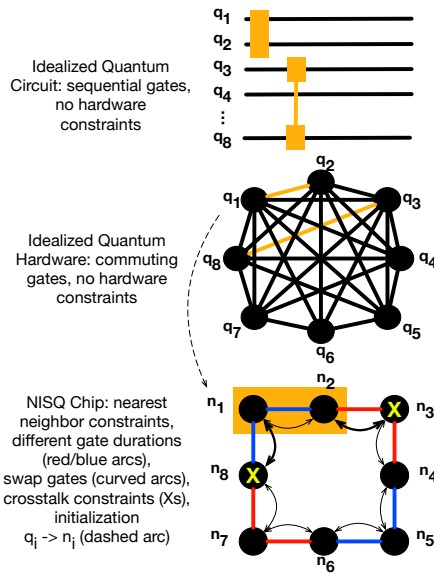

Figure 1: Pictorial view of QCC-NISQ concepts. At the highest level, an *idealized quantum circuit* specifies a sequence of quantum logical gates over *qubit states* that solves a specified problem (top). The idealized quantum circuit could conceptually be implemented on fully-connected quantum hardware in which gates can be carried out between all pairs of *physical qubits*, which is depicted here by a fully-connected graph. Qubit states in an idealized quantum circuit are mapped onto physical qubits in the fully-connected architecture. At this level, gates are specified between physical qubits and can be executed in parallel if they do not involve the same qubit (middle). In an actual NISQ chip, physical gates can only be carried out between a subset of pairs of qubits, usually nearest neighbors in a 1D or 2D array. To carry out 2-qubit gates specified in idealized quantum circuits between qubits that are not connected, *swap* gates are added to route logical qubit states to physical qubits that are connected so that the desired gates can be applied (bottom).

critical to minimize the duration of compiled circuits. Thus, compilation is challenging due to: the parallel execution of gates with different durations, the planar or quasi-planar topology of the qubit locations on the chip, the ordering constraints from the original idealized circuit, as well as additional constraints such as cross-talk.

**Example:** Figure 1 shows a concrete QCC example requiring gate operations $g_A = \mathcal{G}(q_1, q_2)$ and $g_B = \mathcal{G}(q_3, q_8)$. At the top, the algorithm is specified, as is typically done in the gate-model quantum computing literature, as an idealized quantum circuit, with sequential specifications of 2-qubit gates over qubit states. There are no hardware constraints; further, some ordered pairs of gates in the idealized circuit may *commute* (i.e., could execute in arbitrary order, even simultaneously, and still produce correct results). A fully-connected quantum hardware, with no hardware constraints

except that two gates involving the same qubit cannot be carried out at the same time, is represented in the middle as a complete graph connecting all possible pairs of qubits. The gates $g_A$ and $g_B$ are indicated as a subset of this graph (yellow edges). A corresponding real-world NISQ chip has gates between only a small subset of qubit pairs (bottom). For instance, the gates acting respectively on qubit states $q_1$ and $q_2$ and on $q_3$ and $q_8$ can be executed, even concurrently, when these states are located at pairs qubits connected in the chip. Furthermore, on the actual chip, gate execution durations may differ depending on the actual location they are executed, indicated by red or blue edges in the chip (bottom). Cross-talk constraints preclude operations on qubits that are physically located nearby an activated gate. A 2-qubit gate operating on the quantum state residing on qubit $n_1$ and $n_2$ will prevent any other gate operating concurrently on $n_3$ or $n_8$; this is shown by the yellow Xs (bottom). In this example, we assume the initial assignment of quantum states to qubits allocates $q_i$ to qubit $n_i$, shown as a dashed line. We see that $g_A$ can be applied immediately, but $g_B$ involves two qubit states that are not mapped to nearest neighbors. One solution is for the QCC software to add additional "swap" gates. For instance, swap($n_1, n_8$) and concurrently swap($n_2, n_3$), shown in bold (bottom), will bring the states $q_3$ and $q_8$ to qubits where $g_B$ can be applied. This highlights that the QCC procedure can also determine the *initial assignment* task (i.e., decide the initial qubit location for each qubit state on the idealized hardware) to optimize the gate schedule. For example, the QCC solver could decide to initialize $q_3$ and $q_8$ on two adjacent qubits on the actual hardware chip, avoiding all swaps for executing the idealized circuit in question.

To summarize, to compile from the idealized quantum circuit illustrated at the top of Figure 1 consisting of two gates $\{\mathcal{G}(q_1, q_2), \mathcal{G}(q_3, q_8)\}$ and no hardware constraints, into the sequence of (parallel) gates that can be executed in the actual NISQ hardware chip at the bottom of Figure 1, the QCC solver will: (1) first find the initial locations for the four qubit states $q_1$, $q_2$, $q_3$, and $q_8$ on the NISQ chip; then (2) add auxiliary swap gates to bring the two pairs $(q_1, q_2)$ and $(q_3, q_8)$ to adjacent physical qubits that are connected and execute the required operations; and (3) schedule all the gates, each with possibly different duration, to execute in parallel in the shortest amount of time, while obeying all hardware constraints such as cross-talk constraints.

**Existing Work on QCC:** Since the development of NISQ superconducting processors, there has been development of software libraries to synthesize and compile quantum circuits from algorithm specifications (Wecker and Svore 2014; Smith, Curtis, and Zeng 2016; Steiger, Häner, and Troyer 2018; Barends and others 2016), including work explicitly addressing theoretical bounds on the overhead introduced by swap gates (Beals and others 2013; Brierley 2017; Bremner, Montanaro, and Shepherd 2017). Recently, it was proven that the QCC problem and its common variants are NP-Complete (Botea, Kishimoto, and Marinescu 2018). (Bhattacharjee and Chattopadhyay 2017) investigated approaches with off-the-shelf MILP solvers, such as Gurobi, for solving QCC. (Guerreschi and Park 2018; Zulehner and Wille

2018) developed heuristics that address NISQ constraints while minimizing the number of required swap gates. The initial assignment task has been recently addressed in (Paler 2019). Policies for compilation in chips with variable performance parameters among different qubits and gates have been studied in (Tannu and Qureshi 2018), as well as in (Murali et al. 2019). In (Oddi and Rasconi 2018; Rasconi and Oddi 2019), the authors present heuristics (greedy randomized search, and genetic algorithms) specifically designed to solve the QCC Max-Cut benchmark set that was introduced in (Venturelli et al. 2018). In (Booth et al. 2018), CP is explored as an alternative and complementary approach to the temporal planning methods introduced in (Venturelli et al. 2018). Other methods are proposed in (Li, Ding, and Xie 2018) and in (Childs, Schoute, and Unsal 2019), where the crosstalk-free compilation of circuits is handled heuristically on benchmarks for the IBM chip. In (Khatri et al. 2018), an approach based on iteratively learning a sequence of native gate implementing a target unitary is introduced, solving the problem of compilation with that of gate-synthesis at the same time, for small shallow circuits. Another recent learning approach in (Jones and Benjamin 2018) converts the approximate compilation problem into an auxiliary quantum variational algorithm native on the hardware. In (Nash, Gheorghiu, and Michele 2019) and (Kissinger and Meijer-van de Griend 2019) the QCC problem is solved heuristically for circuits consisting of CNOT gates only.

## 3 Automated Reasoning for QCC-NISQ

Our approach to solve the QCC-NISQ problem is to deploy a general-purpose software suite that leverages *model-based* approaches, including *temporal planning* and *constraint programming* (CP) for the solution of the actual combinatorial problem. Since quantum computing is still a relatively new paradigm, different quantum hardware architectures and algorithms running on them are introduced and revised frequently, with no current clear winner. Therefore, instead of a machine-specific QCC tool, there is a tremendous benefit in developing a general-purpose QCC software suite that is capable of addressing different hardware architectures, different quantum algorithms running on them, and different optimization problems that can be solved by those algorithms. Furthermore, our approach is a very attractive option because: (1) declarative model adjustment (e.g., through a declarative planning model) can adapt quickly to revised hardware designs and constraints, and can also cover a wide range of hardware architectures; (2) existing hardware from various companies are still limited in size, thus even general-purpose algorithms can quickly find good, sometimes proven optimal, solutions.

**Why AI Planning?** In planning, a planner searches for a set of actions that can be executed in sequence to achieve the pre-defined set of goals, while satisfying all domain constraints. In model-based temporal planning, specifically PDDL-based planning, actions can have different durations and can be executed in parallel. We choose planning to be the center piece of the QCC solver suite because:

- The action model and plan constraints in PDDL planning can be used to describe gate operation naturally, capturing chip layout, gate duration, and domain constraints such as "cross-talk". This flexibility lets us model various chips from all of the groups mentioned above developing NISQ chips.

- Any new adjustment or hardware updates from the manufacturer can also be reflected easily in planning model updates, without extensive writing of new software.

- The default objective function of optimizing makespan for most temporal planners fits well with quantum decoherence (discussed in the previous section).

- There are multiple off-the-shelf open-sourced PDDL temporal planners that have been tested through multiple International Planning Competitions (IPC), providing a rich set of algorithms, ranging from exact to anytime, to test on the NISQ-QCC problem.

For more details on the AI planning approach to NISQ-QCC, see (Venturelli et al. 2018).

**Why Constraint Programming?** Constraint Programming (CP) is a paradigm for modeling and solving combinatorial optimization problems, leveraging a diverse set of techniques from fields including operations research and artificial intelligence (Rossi, Van Beek, and Walsh 2006). CP is more general than other discrete optimization paradigms, such as integer programming, as it allows variable types beyond integer and continuous (e.g., interval and set variables), and drops the linearity on the constraints and objective function. CP complements PDDL-based planning for the QCC software suite for a number of reasons, including:

- The scheduling characteristics of QCC (i.e., gate durations, precedence constraints, makespan minimization, and unary qubit capacity) are readily modeled within modern CP solver software.

- Solutions found by PDDL-based planning or a heuristic method can be used as a starting point for the CP search, leading to subsequently better solutions.

- The CP search is an exact algorithm, ensuring that, given enough runtime, the optimal quantum circuit compilation will be found. Additionally, modern CP solvers provide anytime bounds on solution quality in the form of an optimality gap.

For more details on the CP approach to optimize circuits together with planners, see (Booth et al. 2018).

## 4    System Architecture

The objective of the whole framework is to provide to quantum computing researchers the ability to efficiently deploy executables of specific quantum algorithms by keeping control over the tradeoffs imposed by the different choices related to what strategy to adopt to solve the QCC-NISQ problem. While the key component of our suite is the QCC solver that does the compilation, multiple additional tools complement the software suite.

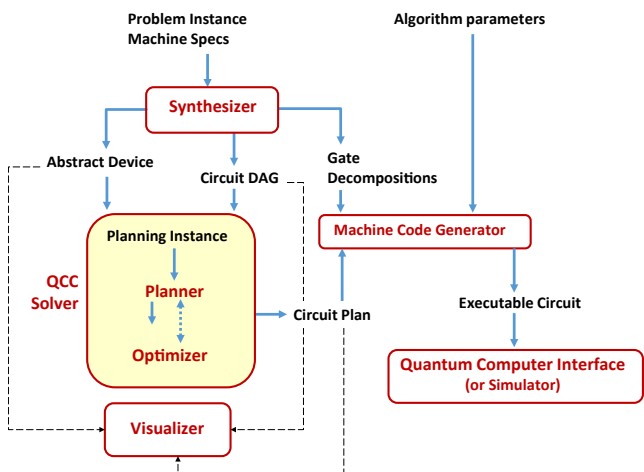

Figure 2: The software suite architecture. Arrows indicate input-output relationship between data structures (black text) and software components (red text). The visualizer can be optionally available (dashed lines) to display the data structure that interacts with the QCC solver in a compelling way. See text of section 4 for details.

Figure 2 shows the architecture of our software suite, with the initial inputs provided by the users consisting of two main components: the *problem instance* containing details on the problem for which we attempt to solve using a quantum algorithm, and the *machine configuration* specifying the details of the gate-model quantum chip (e.g., physical layout, gate durations, constraints). (A third input, the *algorithm parameters*, concerns the setting of the classical parameters of the gates once they are already compiled, and will be discussed in connection with the generation of the executable.)

These inputs are provided to the *Synthesizer* that generates three different outputs (described in later sections), that can be fed into two other components: the *QCC Solver* and the *Machine Code Generator (MCG)*. The *QCC Solver* generates planning problems in the standard Planning Domain Definition Language (PDDL) and generates plans in the standard IPC format, which can then be parsed in a *circuit plan* representation or fed into the CP solver (the "optimizer") to generate a better quality solution.

The last two components are: the *Visualizer*, which can show graphically the machine logical layout, the high-level problem specifications, and the QCC solutions; and the *Machine Code Generator*, which maps the QCC solution and information on how to synthesize gates onto a particular hardware architecture to produce the machine-specific executable file (e.g. a python script) that can submit a job on the cloud to a quantum NISQ device through its public API.

In the rest of this section, we will describe in more detail the different components and their inputs and outputs.

**(Circuit) Synthesizer (CS):** the CS is in charge of two main tasks: (1) generate the low-level gate synthesis (*gate decom-*

*positions*) to act as one input to the MCG; and (2) generate the inputs for the QCC Solver (*circuit DAG* and *abstract device*).

The first task is meant to decompose the abstract gates, which appear in the idealized circuit of the quantum algorithm, into elementary gates supported by the hardware, whose duration in nanoseconds is known at all possible locations in the chip. This decomposition is known as the *gate-synthesis* problem. It is non-trivial in the general case, but for many quantum algorithms and for standard universal elementary gate sets, optimal decompositions are known. Currently, the CS implementation just consists of a lookup library of known decompositions into elementary gates [1]. This decomposition library has been found by various methods, including proven optimal results (Vatan and Williams 2004). The CS assigns a total duration (in standardized clock units) to each possible logical gate, to inform the second component of the module that will have to generate the input files to the QCC solver.

The second task is to instantiate an *abstract device* software object, which is representing the topology of the hardware but is aware only of the different types of gates that need to be scheduled (including swaps), which are represented as edges, and their duration obtained through synthesis (the edge weight). The abstract device includes information about crosstalks/simultaneity constraints in the form of an extra graph whose vertices are the edges of the hardware graph and whose edges indicate impossibility of concurrent gate operations using the corresponding edges of the hardware graph. Fig. 1 (bottom) is a pictorial representation of some informations contained in the *abstract device*. While the properties described in the Abstract Device instance are tailored to the QCC solver capabilities, the device class should be used by different solvers. For instance, if a non-temporal method is used as a solver instead of temporal planning, the gate durations could be discarded.

Finally, the CS composes a Directed Acyclic Graph (Circuit *DAG*) representation of the problem instance, which takes care of defining the partial ordering rules of the gates composing the circuit. The vertices of the DAG correspond to gates on specified qubits and the arcs correspond to precedence constraints; operations that are incomparable by this relation can be scheduled in any order relative to each other. This freedom arises naturally in quantum computation for quantum gates that "commute" with each other, i.e., produce the same effect regardless of the order in which they are applied. But can also be used in the context of heuristic algorithms that try to balance the effectiveness of the logical circuit and cost of implementing it as a physical circuit. The DAG includes only the synthesized two-qubit gates that are necessary for the logical description of the algorithm.

**QCC Solver:** The QCC solver takes the SC input and produces internally the PDDL files that represent the *planning instances*. The ultimate objective of this module is to output the *circuit plan*, which is the compiled representation of the target algorithm. The following options are currently implemented:

- *With/without cross-talk constraints:* specifying whether or not the underlying hardware has cross-talk constraints between adjacent qubits; see Figure 1 (bottom).

- *With/without qubit initialization:* specifying whether or not the QCC Solver should decide (as part of the planning objective) the mapping of specific qubits to quantum states at the beginning of the algorithm; see Figure 1 (dashed line, middle-bottom).

- *Single or multiple phases:* specifying if the idealized circuit should be executed multiple times in sequence (see Section 2). This is particularly important for QAOA circuits, which require insertion of alternating "phase-separation" and "mixing" sets of gates to increase the accuracy of the solution returned by the algorithm. Requirements on running multiple phases will lead to CS generating PDDL files with different sets of actions and goals.

The planning instance is processed by a *Planner* to obtain a temporal plan, which is a sequence of gates to be executed on the designated hardware architecture. Specifically, at the moment we use two different macro-approaches to generate the final plan[2]:

- Use off-the-shelf temporal planners that can take the standard PDDL input. The following planners have been used: LPG, TFD, POPF, CPT, and SGPlan; all previous winners at different IPC. The performance of different individual planners is reported on in (Venturelli et al. 2018).

- Use a combination of planning and CP in a hybrid setting where plans found by any temporal planner can then be used to warm-start (i.e., seed) the CP model that in turn is solved by the commercial software CP Optimizer to find a new, better quality plan. The evaluation of this approach is described in (Booth et al. 2018) [3].

Results on the use of the QCC solver with the various variants for MaxCut QAOA are presented in (Venturelli et al. 2018) and (Booth et al. 2018). The QCC solvers have been also configured for tests on Graph Coloring, which are currently being performed. The versatility of our approach utilizing off-the-shelf domain-independent PDDL temporal planners is reflected in the ability to quickly test machine models from different companies (Rigetti, Google, IBM) at different scales and diffferent chip layouts, for different set of gate operations, gate durations, and gate constraints.

Figure 3 shows the visualizer displaying a single phase of QAOA to solve a Maxcut problem (referred to below as Maxcut-QAOA) on the Google Bristlecone chip. The figure shows the following components:

- *Goal Graph:* at the top-left corner, the goal graph shows the gates to be scheduled (colored edges) according to the

---

[1]Elementary gates might include CZ, CNOT, iSWAP, the PhasedXPowGate of Google's chip and single qubit rotation gates.

[2]The plans generated either by planner or CP Optimizer are validated by the official plan validator software VAL before passing on to the next component.

[3]The CP model is not automatically generated.

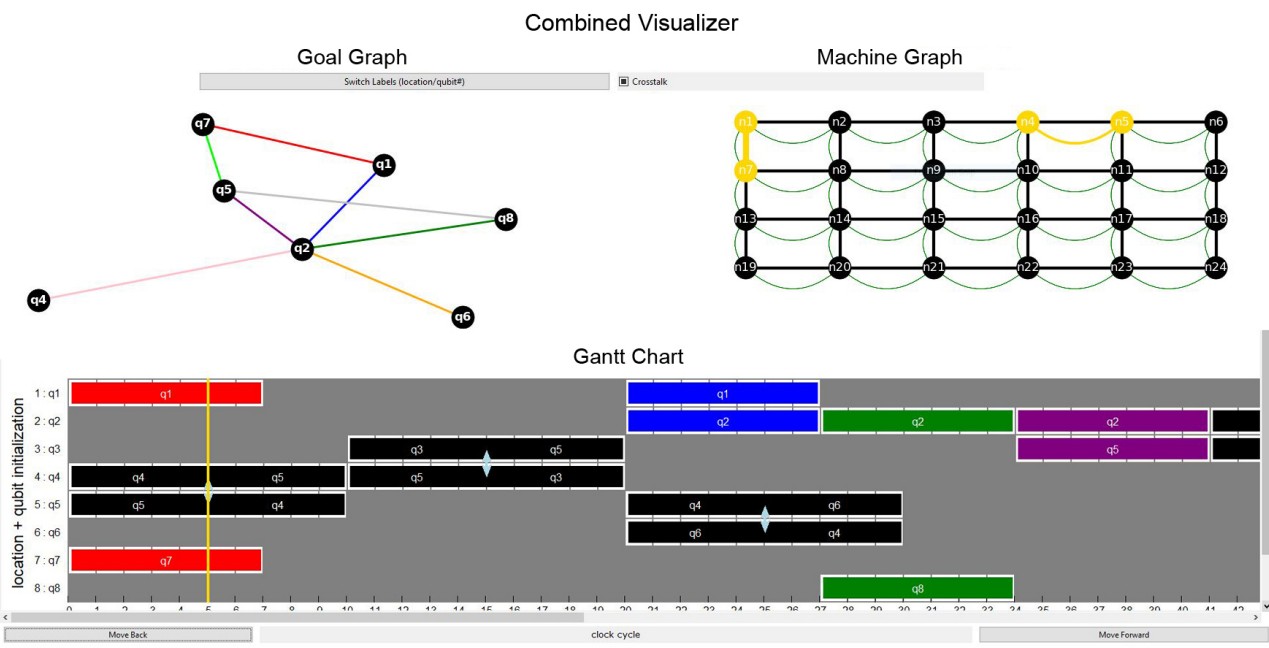

Figure 3: Our visualizer interface with: Goal Graph (top-left), Machine Graph (top-right), and QCC plan (bottom).

idealized quantum circuit. For Maxcut-QAOA, the goal graph is identical to the actual graph that we want to cut. The goal graph contains essentially information related to the *circuit DAG* (Fig. 2).

- *Machine Graph:* at the top-right corner, the machine graph represents the underlying logical layout of the NISQ processor target (in this case, a portion of Google's Bristlecone chip). Multiple connections between a given pair of qubits represent different types of gates that can be applied to the nearest-neighbor connected qubits. Each type of gate has a different duration. The machine graph contains essentially information related to the *abstract device* (Fig. 2).

- *QCC plan:* this shows the temporal schedule of gate operations as a Gantt chart, each with its starting time, duration, and the qubits involved in that particular gate operation. The gates are color coded to match the high-level goals (see the matching edge color on the *Goal Graph* described above). Sliding over the timeline of this Gantt-chart representation will also highlight on the machine graph the qubit set and the active gates operating on that set, at the timepoint selected on the timeline of the plan. For reference, in Figure 3, the slider has been set on time slot number 5. The QCC plan contains essentially the same information related to the circuit plan (Fig. 2).

For the case of graph coloring, the QAOA algorithm logical gates require multiple kind of two qubit gates, as explained in (Hadfield et al. 2019). A more advanced visualizer that is able to show the various steps by using different colors, graphic notations and animations is under development.

**Machine Code Generator (MCG):** this component outputs the machine instructions that perform the compiled algorithm on the target quantum processor or in a simulator (the *Executable circuit*). The MCG integrates the abstract QCC solution (the *circuit plan*) with the gate synthesis instructions generated by the CS, and sets the parameters that are required for the execution of the algorithm, which are given as inputs (*Algorithm parameters*). More specifically, it unwinds the duration abstraction of the gates that has been scheduled and replaces the composite gate with the code that activates the exact sequence of native gates on the chip with the correct parameters. For instance, for the case of QAOA, these includes the angles for the alternating unitary transformations that compose the algorithm. The algorithm specifications not only set the individual gate parameters, but also many control-loop policies for hybrid classical-quantum computation. Examples include the measurement and accuracy evaluation functions that determine whether the algorithm needs to be iterated one more time with different parameters after execution.

**Current deployment and implementation details:** The software suite we describe ultimately will be integrated with the general software framework for programming quantum processors QuaSar (NASA's *Quantum user-assisting Software for applied research*. QuaSar is a high-level backend and frontend system, soon to be released, which support transparent inter-operability between code written for most quantum computers that released APIs, including Cirq, Rigetti Computing PyQuil as well as QASM 2.0.

The current implementation of the CS, and of the QCC Solver is in Cirq (Google AI 2018)[4], an opensource framework that conveniently abstracts several aspects of temporal manipulation of operations in quantum circuits. This excludes the actual implementation of the planner and the CP optimizer, that varies case by case (they are considered external black boxes to be interfaced). The output of the MCG is a Cirq schedule object; such an object can be run directly on Google's hardware or converted by QuaSar to the common quantum circuit format QASM for compatibility with other hardware providers.

**Relation to existing work on software frameworks:** as reviewed in (LaRose 2019), most quantum computing companies have released software frameworks that allow end-to-end deployment of quantum algorithms. These packages could be as simple as generic wrappers around machine instruction languages, or include suites for handling specific aspects. For instance, Xanadu's PennyLane focuses on facilitating aspects of development of (quantum) machine learning algorithms (Bergholm et al. 2018), Microsoft's Quantum Development Kit (Svore et al. 2018) focuses on resource estimation for fault-tolerant quantum computers, and Zapata's algo2qpu (Sim et al. 2018) focuses attention on the hybridization of variational algorithms with classical optimization techniques.

Our framework focuses on facilitating the optimization of the compilation by allowing the user to use alternative compilation strategies, possibly hybridizing them, and allowing the inspection of the results of the compilation.

## 5 Conclusions and Future Work

We have introduced a framework for applying AI Planning to the QCC-NISQ problem, and developed a software suite that implements components of that framework. The framework and suite are completely general and can be used to target quantum computing hardware devices of different types and by different companies. Different planning algorithms, complemented by hybrid algorithms using Constraint Programming, can be used to compile a qunatum circuit to a specified hardware device. The compiled circuit can be visualized, enabling users to understand how the compilation of the circuit interacts with the constraints of the device. The resulting compiled circuit can then be run on actual hardware, or simulators therof, by different companies. The flexibility of declarative AI planning models allows us to solve the QCC-NISQ problem for these diverse hardware architectures, and to evolve our solutions as the details of the hardware evolve.

The suite integrates several planners (LPG, POPF, TFD, CPT, and SGPlan), together with IBM's CP Optimizer suite, the best performing commercial software of its kind. We have generated tens of thousands of instances of the QCC-NISQ problem based on pairing QAOA for MaxCut on random graphs with quantum computer architectures of varying size and constraints. Performance comparisons of different planners and hybrid approaches are reported in (Booth et

al. 2018; Venturelli et al. 2018) and benchmark instances are available online[5] and have already been used to test other QCC-NISQ solver approaches (e.g. in (Oddi and Rasconi 2018)). Future hybrid efforts are under development, including the use of different planners hybridized with LPG, which can take as input a "seed" plan. While CP improves on seed plans, it requires a separate modeling effort; hybridizing multiple planners uses multiple algorithms but requires no additional modeling effort.

We claim that a general-purpose software suite built on declarative planning algorithms and constraint programming is a promising approach to addressing the constraints of NISQ devices. A highly optimized problem-specific tool may perform better in some cases, but at the cost of significant engineering effort that cannot be reused for new problems. Our model-based, automated reasoning approach is very flexible with respect to features of the hardware graph, including irregular structures, as often arise from manufacturing imperfections. The ease and expressiveness of PDDL modeling facilitates the inclusion of additional features that are characteristic of quantum computer architectures, such as the ability to *quantum teleport* quantum states across the chip (Copsey et al. 2003), providing more flexibility than mere nearest-neighbor swap.

The modularity of the software suite we introduced allows it to be improved iteratively, giving a foundation for future work on the application artificial intelligence methods to quantum computing. Here, we focused on the makespan of the compiled circuit as the objective function to minimize, but data from ongoing experimental work will likely yield more sophisticated quantities to optimize. We are also actively looking at compiling circuits corresponding to QAOA for different combinatioral optimization problems (e.g. job shop scheduling), many of which involve new types of multi-qubit gates with different characteristics from the simple ones used in illustrative examples in this paper and in other introductory publications. Another avenue of extension of our work is to generalize the NISQ-QCC problem and our software suite to the specificities of quantum processors that are dramatically different than superconducting NISQ devices. For instance, soon to made available Iontrap processors such as the one of IonQ (Nam et al. 2019) or of Honeywell International Inc. feature a number fully-connected cells of qubits (where qubits interact through a different set of native gates than the ones of superconducting processors) which are in communinication to each other through the ability to swap quantum information via photonic interfaces. In a future work, we intend to provide examples for different architectures including sample code. We plan to make the architecture available (to be interfaced with a planner and optionally a CP optimizer) as an opensource package.

Finally we believe that our approach should be of great interest to the community developing low-level quantum compilers for generic architectures (Steiger, Häner, and Troyer 2018; Häner et al. 2018),

---

[4]https://github.com/quantumlib/Cirq

[5]https://ti.arc.nasa.gov/m/groups/asr/planning-and-scheduling/QCC_ICAPS18.zip.

to designers of machine-instructions languages for quantum computing (Smith, Curtis, and Zeng 2016; Bishop 2017), and to developers of unifying frameworks for quantum computing software toolchains and interface to solvers (McCaskey et al. 2018).

**Acknowledgement:** The authors would like to acknowledge support from the NASA Advanced Exploration Systems program, NASA Academic Mission Services (NNA16BD14C) and the NASA Ames Research Center. B.O. was supported by a NASA Space Technology Research Fellowship.

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
