# OpenReview forum: "Quantum Circuit Compilation: An Emerging Application for Automated Reasoning"
_icaps-conference.org/ICAPS/2019/Workshop/SPARK — SPARK 2019_

### Official Review · AnonReviewer1 · 2019-04-26
**A very interesting topic, described in a fairly complete description of an innovative software framework that tackles the quantum circuit compilation problem.**

**Rating:** 4
**Confidence:** 3

**Review:**

This paper presents a software framework that supports a general automated reasoning approach relatively to the quantum circuit compilation problem (QCC).

This work is significantly based on the results proposed in two recently published papers in the QCC literature, namely VenturelliEtAl2018 and BoothEtAl2018, leveraging such results and presenting a software suite architecture in which the analyses performed in those papers can be replicated, but can also be generalized to different cases. The main contribution of the paper is therefore the previously mentioned architecture, which is intended to help the human expert in all the phases of quantum problem compilation problem, from the specification of the hardware quantum machine, to the production of the executable circuit ready to be submitted to real (or simulated) quantum hardware. The proposed architecture is also equipped with a visualizer that allows the graphical representation of the obtained intermediate results.

This work is indeed valuable as it tackles a very interesting problem, widely analyzed in recent literature, which is becoming a hotter and hotter topic. The paper is also valuable in that it presents to the non-knowledgeable reader a good introductory presentation of the whole problem, as well as a rather complete literature review.

The reviewer understands that QCC problems that can be possibly tackled using the presented framework, but other than those analyzed in the two papers of major reference, are still at a preliminary stage of analysis; yet, this is not so important: this work can be of great interest for the SPARK audience, and should therefore be presented.

One main suggestion is the following: this paper does not present any strictly technical detail as of why the approach that leverages temporal PDDL-based planners, or the hybrid PDDL+CP approach, guarantees the level of generality and versatility promised by the proposed framework. This is understandable, given that the focus of the paper is on the framework and not on the strictly technical aspects of the involved declarative paradigms. Yet, some more details that demonstrate the fesibility and the scalability of the selected approach should be added to the workshop presentation, in case the paper is accepted. My opinion is that this paper is an accept.

---

### Official Review · AnonReviewer2 · 2019-04-30
**Does SPARK need another paper about QCC?**

**Rating:** 2
**Confidence:** 2

**Review:**

This paper follows previous papers by the authors on formulating the quantum circuit compilation (QCC) problem as a temporal planning or constraint-based scheduling problem.

While the paper may serve as an introduction to the topic for readers who have not seen the previous papers, I found myself asking what, if anything, is new in this paper, and specifically what is new that relates to the planning/scheduling problem that is formulated or how it is solved. In the introduction, the authors state that this paper reports on "our software suite in its entirety: its components, engineering, and deployment ...". First, I did not find much said about deployment, only that the system will be integrated with a framework developed/being developed at NASA, and also not much about the engineering of the software system. E.g., were there any particular software engineering challenges, e.g., related to design, performance, quality, etc, in the construction of this system (not in solving the planning problem; that has, as the authors say, been reported elsewhere) that are not usually encountered in other systems that have a planning-based component inside them? If there were, I did not find anything said about them in the paper.

But more importantly, does the overview of the components of this system that the paper presents tell us anything new about the planning problem? Does it point at ways that the problem can be decomposed or modeled differently, with the planner taking a greater or smaller part of the whole QCC chain? Does it analyze or demonstrate why the problem decomposition that is used in the current system is the right one? Does it analyze or evaluate the way that this system interacts with its hypothetical users? I did not find any such questions posed or answered in the paper. It describes the problem and the way that planning is used in solving it at a level that is no more informative than the previous ICAPS publications about the system.

---

### Official Review · AnonReviewer3 · 2019-04-30
**Review3**

**Rating:** 5
**Confidence:** 2

**Review:**

The paper describes a general software suite for deploying executables of quantum algorithms to specific hardware.  It leverages both AI Planning and Constraint Programming to form a general solution that allows the user to specify the hardware architecture and algorithm, which is then automatically compiled to the hardware.  A GUI shows the goal graph, machine graph and Quantum Circuit Compilation in a format that can aid the user in understanding the way in which the algorithm is running on the hardware.   A key benefit of this software suite is its generality due to its use of general problem solving techniques, which means that the same software can be applied to many different algorithms and, perhaps more importantly, many different hardware types.

The paper is easy to read, clearly well motivated, and generally of high quality.  The paper would be a great addition to the SPARK workshop.

My only minor suggestion is to switch to an Author (Year) style of citation when a citation refers to a work.  For example, prefer "Author et al. (YEAR) discuss ..." to "(Author et al., YEAR) discuss".

p3: cross-talks constraints -> cross-talk constraints

---

### Decision · Program_Chairs · 2019-05-08
**Acceptance Decision**

**Decision:**

Accept

**Comment:**

Relevant and interesting for SPARK